# How a Transformation towards Sustainable Community Catering Can Succeed

**Viviana Lopez, Jenny Teufel * and Carl-Otto Gensch**

Oeko-Institut e.V., Merzhauser Straße 173, D-79100 Freiburg, Germany; v.lopez@oeko.de (V.L.); c.gensch@oeko.de (C.G.)

**\*** Correspondence: j.teufel@oeko.de; Tel.: +49-761-45295-252

**Abstract:** Community catering or to use another common term especially in the American literature institutional foodservice plays a central role in changing our food system towards sustainability. Community catering establishments can bring about changes in this context at various levels. Hence, in the context of menu planning, they have a direct influence on the level of meat consumption. Indirectly, however, they can also support changes in eating habits by offering the guest an equally attractive alternative, thus giving him or her a sense of how tasty a low-meat cuisine can be. On the basis of this experience, the consumer may possibly change in turn his or her own purchasing behavior and menu planning at home. With the increasing importance of catering for day-care centers and schools, community catering also has a considerable influence on the nutritional status as well as on the development of people's individual diet and the later eating habits of young people. By understanding socio-technical systems as embedded in ecological systems this paper takes a systemic view on innovations in transformation domains as the objects of desire for governance towards sustainability. The framework developed in the context of the BMBF-funded research project "Governance model for socio-ecological transformation processes in practice: development and testing in three areas of application" known by its acronym TRAFO 3.0 was applied to examine innovative approaches and actors in community catering and their contributions to more sustainable food systems. A number of studies show that a very large environmental relief potential can be achieved by reducing the quantity of meat and other animal products offered. However, the concrete implementation of this goal is associated with a multitude of challenges, since meat-containing meals are an important part of German food culture. How the transformation towards meals with fewer animal products in German community catering can succeed is an important question in the context of the transformation to sustainable food systems. To answer this question, we analyzed the status quo of the socio-technical system of German community catering using a developed governance model. One of the central results was that community catering stakeholders who have successfully reduced their offer of animal products died fundamental changes in meal planning. Cooks had to "reinvent" meals completely to be successful.

**Keywords:** community catering; institutional foodservice; sustainability transformation; transformation governance; sustainable food systems; meat reduction; menu planning; social innovations; environmental policy

## 1. Introduction

The average dietary style in Germany contains—compared to the recommendations of the German Nutrition Society (Deutsche Gesellschaft für Ernährung—DGE)—too much meat, too many saturated fatty acids, too much sugar and not enough vegetables. The consequences are obesity and diet-related diseases. The DGE suggests a reduction of the average weekly meat consumption from at present

1000 g (men) and/or 600 g (women) to a maximum of 300 to 600 g per week and a tripling of the consumption of vegetables from a health risk point of view [1].

In addition, dietary styles based on a high proportion of meat and other animal products are associated with multiple negative environmental impacts, including a high greenhouse gas emission potential, high land use requirements and a high nitrogen emission potential. If one considers various food product groups in relation to nutritional needs, meat and meat products are associated with the greatest environmental impacts [2–4]. Compared to the statistical average diet, the conversion to a lower-meat, vegetarian or vegan diet has been linked to a high potential for reducing such impacts [5–8].

Due to sociodemographic developments and changes in occupational and time structures, the community catering sector or rather the institutional foodservice sector, as a segment of the fast-growing out-of-home catering market, is becoming increasingly important in terms of transforming our food system towards sustainability.

Against the background of the development of the out-of-home food market, possible transformation strategies to achieve sustainable production and consumption patterns in the field of nutrition play an important role [9]. Community catering as a case study within the out-of-home food market is also interesting because, in addition to the direct effect of changing local consumption, there are also indirect effects in that consumers may try out new menus and possibly also change their diet at home. With the increasing importance of catering for day-care centers and schools [10], community catering also has a considerable influence on health, nutritional habits, as well as on the development of one's own nutrition style and the subsequent food consumption behavior of young people.

However, the shift towards a low meat diet or a diet with a high proportion of plant-based ingredients in out-of-home catering (including community catering) cannot be taken for granted. In Germany, for example, curry sausage with French fries and Wiener Schnitzel are still among the most popular cafeteria meals in most schools, universities, public institutions and hospitals.

The implementation of concrete strategies aimed to reduce meat consumption through community catering have proved to be associated with a number of complex challenges and unintended consequences such as those shown by the failure of the request for a mandatory "Veggie Day" in Germany which was promoted by the Bündnis90/Die Grünen parliamentary group in 2013 [11,12].

The challenges and obstacles associated with the transformation and the governance approaches they convey are being investigated within the framework of the BMBF-funded research project "Governance model for socio-ecological transformation processes in practice: development and testing in three areas of application" known by its acronym TRAFO 3.0 (http://trafo-3-0.de).

## 2. Theoretical Framework

### 2.1. From Transitions to Transformation Pathways

As starting point in understanding change processes in current systems with the aim of moving towards sustainability, concepts from transitions literature must be recalled. Sustainability transitions are *"processes of fundamental social change in response to societal challenges [ … ] perceived as social problems"* [13]. The persistence of such problems is derived from path dependency of dominant practices and structures which are defined as 'regimes' and whose resolution requires structural and long-term change [13,14]. In this regard Geels [14] highlights that *"addressing persistent environmental problems requires large improvements in environmental efficiency, which in turns requires transitions to new socio-technical systems"* [14], which is why sustainability transitions are inevitably socio-technical.

The concept of regimes is the main element in which transition research has focused to examine socio-technical change since its start. A regime can be defined as *"shared semi-coherent (i.e., relatively stable and aligned) sets of rules or routines directing the behavior of stakeholders on how to produce, regulate and use technologies part of a specific socio-technical system"* [15]. Thus, a socio-technical system is the one which enables the fulfilment of given societal functions under certain regime.

The Multi-level perspective (MLP) [16,17] is one of the most prominent frameworks in transition research [18]. This framework (as represented in Figure 1) differentiates between three analytical levels which evolve with increasing temporal stability: *niche (flexible and fluid), regime (semi-stable) and landscape (slow societal processes that provide the context for regime stability or change).* MLP defines transitions as regime change, and explains its occurrence by an interplay between niche innovation, internal regime change (including resistance) and landscape developments [19]. Along its evolution, this field has also focused its attention on specific actors within particular MLP-levels (e.g., niche and regime stakeholders), thereby attaching more relevance to actions and agency in contributing to understand the roles of these stakeholders in the different phases of a transition process [19].

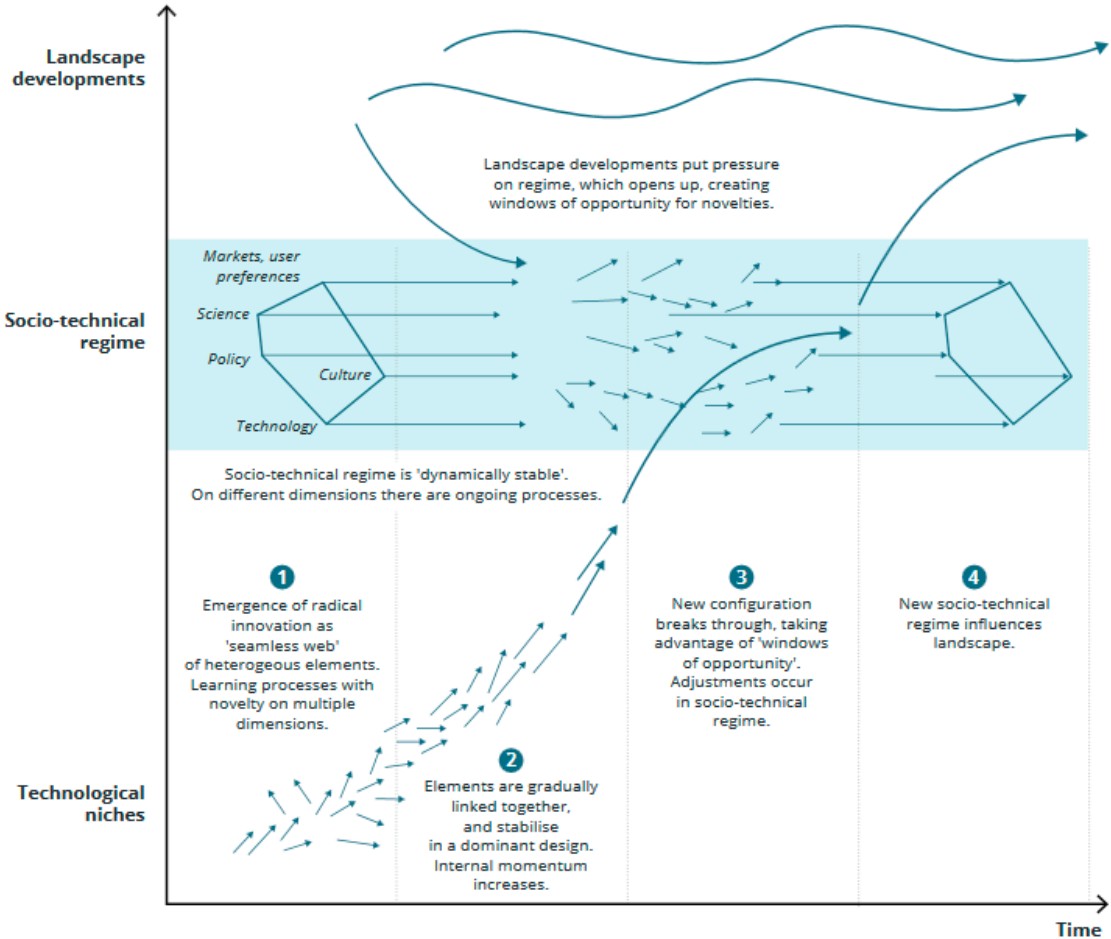

**Figure 1.** Multi-level perspective in socio-technical transitions (source: [16]).

Owing to its origins in innovation studies, innovation is a key concept in transitions literature. In MPL, niche innovations are seen as crucial for socio-technical transitions as these co-evolve along with many other dimensions in diversity of fields such as governance, politics, consumption, civil society, culture, among others [14]. This co-evolution is recognized to influence the acceptance and diffusion of the niche innovations into the stable regime.

In this regard, transitions literature acknowledges that innovation is a social process, which is based on interactions between multiple stakeholders, throughout phases of emergence; diffusion and stabilization. Moreover, understanding that socio-technical systems are the result of human activities implies that transitions are also inherently multi-actor processes involving various social groups with different interests. This is the reason why in enabling socio-technical innovations, cultural and political dimensions have considered to be as relevant as technology and infrastructural developments.

Considering the social nature of innovations, the perception of a socio-technical transition towards sustainability as a top-down process has been challenged from a governance perspective. In this sense, the idea of stakeholders at the niche levels as the "great innovators" was reproduced by other approaches to socio-technical transitions that have put great emphasis in understanding the dynamics of niche formation. The strategic niche management approach (SNM) for example, suggests that local (bottom-up) projects are important in the emergence of niche innovations which are able to create broader outcomes beyond the emergent network [20] and could therefore determine a desired transition.

However, it has been recognized that pathways to a new regime do not always systematically evolve upon the emergence of an innovation niche. In this regard, and arguing that a sole focus on transitions from a regime shift reflects a western bias, Ghosh and Schot [21] advocated for the need of more nuanced ways of understanding and analyzing regime-change without relying on niche innovations. By putting more emphasis on change coming from within the regime, these authors mobilized the concept of *transformation* as one specific pathway of regime change and characterize *optimization* and *transition* pathways through changes in trajectories, rules and selection pressures (Figure 2). Transition, the third pathway, evolves from the three niche-based pathways (reconfiguration and de-alignment and re-alignment) previously defined by Geels and Schot [22].

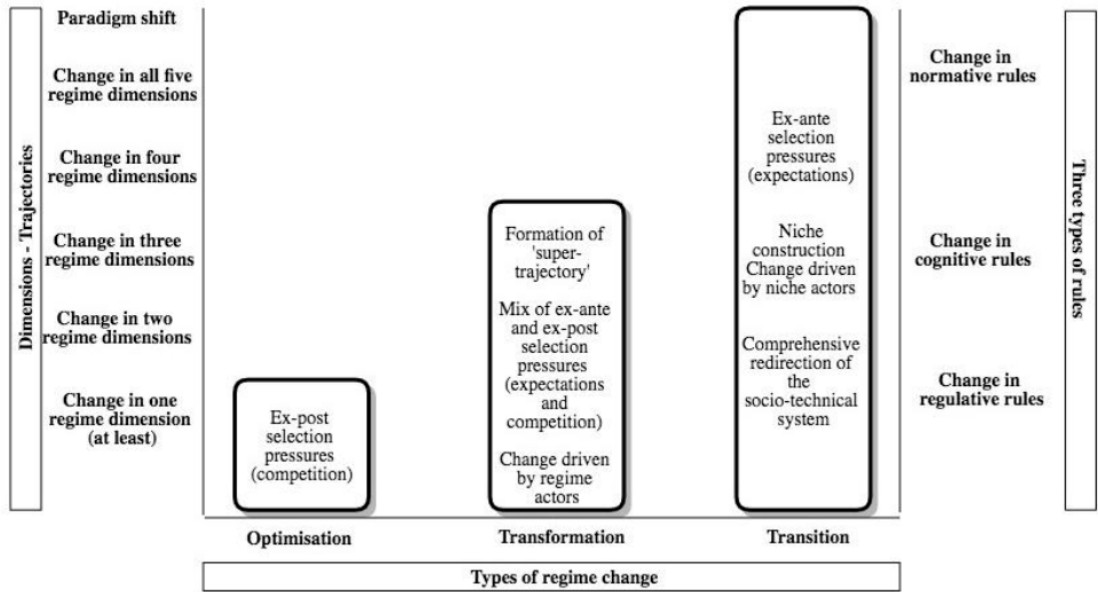

**Figure 2.** Regime change framework (source: [21]).

Following Ghosh and Schot [21] this paper examines changes in socio-technical systems by focusing on actions from regime stakeholders whose actions are considered to contribute to *transformation* pathways. The transformation pathway recognizes that regime stakeholders are often also heavily invested in optimizing, expanding and improving existing systems in order to address sustainability issues [21]. Besides, we believe that this approach contributes to address the dichotomy between agency and system which MPL has failed to capture [19].

### 2.2. A Governance Model for Socio-Ecological Transformation

In order to shift the current energy and resource-intensive lifestyles of our society towards sustainable production and consumption patterns, a sustainability transformation is necessary. This involves a multi-faceted change in societal routines and structures, including the integration of social and technological aspects. Although several theories already exist on the emergence and process of transformations, it still remains unclear how transformations can be strategically promoted, potentially

initiated, and shaped. This was the point of departure of the research project TRAFO 3.0 founded by the Federal Ministry of Education and Research in Germany from April 2015 to September 2018.

Building on theoretical elements of transitions research and evolutionary economics, a draft governance model for examining socio-ecological transformation processes in practice (Figure 3) was developed in the context of the project TRAFO 3.0. This draft model was tested in three fields of application (paperless publishing and reading, wide use of electric bikes, sustainable production and consumption of meat), and the findings from the analysis of these three fields of application were incorporated in an iterative process into the final governance model. Furthermore, based on the results of the three fields of application, a set of approaches for the promotion and governance of small- and medium-range transformations was provided.

In the developed governance model, we understand socio-technical or socio-economic systems, such as energy, nutrition and mobility [23] as regimes which also represent the objects of the desired sustainability transformations. Socio-technical systems are characterized by technologies (e.g., mobility, energy), while socio-economic systems are characterized by markets and functions (e.g., health, education, nutrition). From a governance perspective we may refer to these systems as transformation domains. By taking a systemic approach to the study of transformation domains, we attempt to capture the interrelationship of various actors.

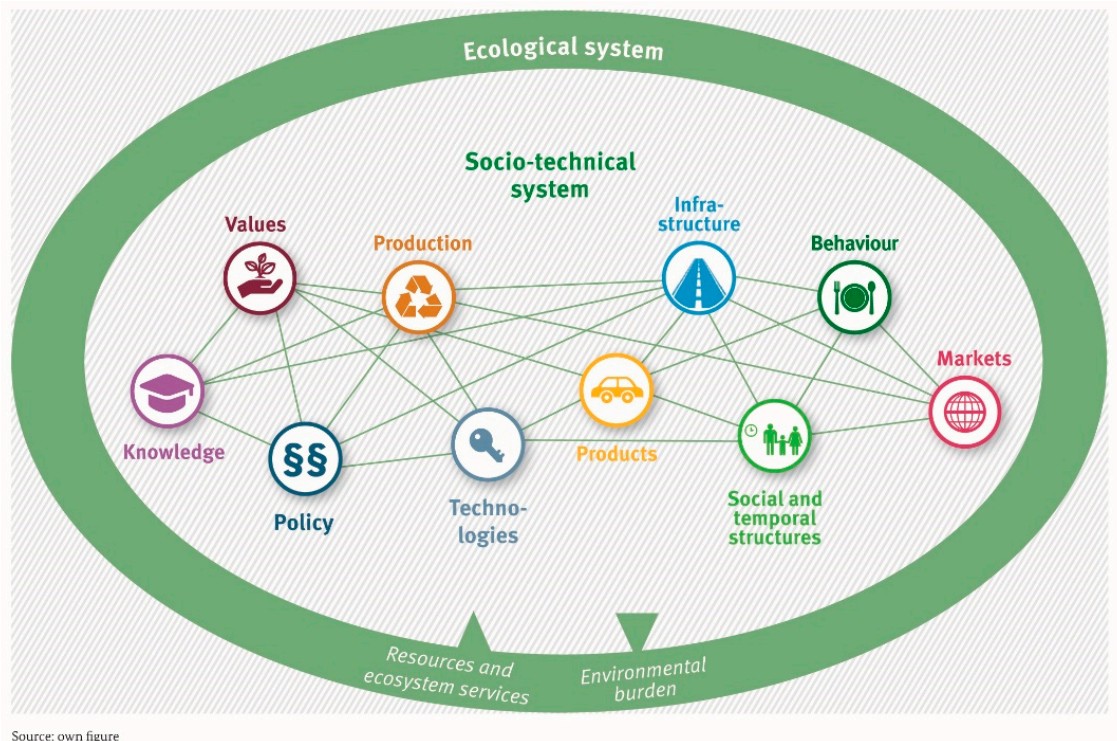

Source: own figure

**Figure 3.** Elements of socio-technical systems and its embeddedness in ecological systems (adapted from [24]).

Behavior determining how we feed ourselves, travel or communicate, for example, is influenced by available products, existing infrastructure and technologies, market configurations and power relations within them. At the same time, all these actors influence and are influenced by normative actors such as values, norms and the policy framework in a given historical moment. These system elements are intertwined and mutually influence one another in determining transformations developments in a given domain.

The environmental compatibility of a specific socio-technical/-economic system is determined by the features and interactions of the various system elements. Socio-technical/-economic systems are embedded in ecological systems as they benefit from their resources and ecosystem services (input)

but also burden them with waste and emissions (output) and are influenced by their individual developments. In this model, transformations represent not only changes to individual elements, but also to the entire system: A societal need is satisfied in a radically different way than in the past. For sustainability transformations the environmental impact should be significantly lower than for the old system. Furthermore, as in transitions literature, transformations can be viewed as the shift from one relatively stable system state (equilibrium) to a different equilibrium.

The intended applicability of this model is on examining different socio-economic systems in order to identify best-practice initiatives and innovative approaches from market actors. The objective is that these findings could contribute to developing transformative environmental policy instruments as a way for governing transformations for sustainable food systems.

Similarly, as those described by transitions literature, the dynamics of transformations encompass different phases. Transformations evolve through preliminary, acceleration and stabilization phases along which specific stakeholders are involved and distinctive challenges arise. Equivalent as the diffusion phase for transitions, the acceleration phase in transformations processes is characterized by a gain in momentum of innovations. Within this phase innovations are increasingly adopted and supported by regime actors and make their way into mainstream markets in order for them to enter and compete or co-evolve with other innovations in the wider socio-technical regime. It is thus acknowledged, that setting the right policy instruments at this point is key in navigating political struggles arising from all stakeholders involved and in promoting the transit to the stabilization phase of a desired transformation.

With this in mind, it is relevant to highlight the need for governance strategies for transformations towards more sustainable systems which are not exclusively aimed at promoting niche innovations. Instead, effective governance for transformation should equally address regimes actors as agents of change along with the variety of elements co-evolving within the regime and landscape levels that would ultimately enable a socio-technical change. From this perspective, we complement the applicability of the above described governance model with Jacob et al. [24] concept of ***transformative environmental policy*** which identifies eight approaches to action:

- Systematic analysis of transformation fields;
- Recognize, evaluate and use societal trends for environmental policy;
- Supporting the development of social mission statements and goals;
- Designing interfaces in and between socio-technical systems;
- Promoting social and institutional innovations and experiments;
- End unsustainable structures (exnovation);
- Involving new stakeholders and stakeholders with new roles;
- Designing policies and processes in a time-sensitive manner.

To complement our approach to regime change, one last concept drawn from evolutionary economics is the understanding of markets as realms for regime transformation. Socio-technical transitions approaches have also outlined how markets act as a selection environment in which products with the highest fit with consumer requirements will survive. As a result, transitions can be seen as change resulting from interactions between variation and selection [14] which are intrinsic processes within market dynamics. Considering this, markets should be the starting point in examining contributions from social innovations to governance for transformations towards a sustainable economy. Existing markets are part of socio-technical systems which have emerged and evolved in order to be able to fulfil a given societal function within established regimes.

Regarding food markets, Sipple and Schanz [25] pose that these are perceived to be hardly accessible for governance approaches and possibilities of coordination for ecological sustainability or regional identity which is why they conclude that by understanding market structures of food supply, related governance approaches and coordination possibilities could be derived. Community catering is one example of a mainstream food market which has been long established to fulfil the

nutrition function in variety of institutional realms (schools, hospitals, government catering, etc.). The accomplishment of such function is interlinked to other markets within socio-technical systems such as agriculture, logistics and food production, which are also subject to being influenced by a variety of stakeholders.

Finally, drawing from the notion that socio-technical changes inevitably involve changes in technologies and changes in markets, cultural meaning, policy and politics [14], concrete aspects related to the system elements from the governance model are to be examined for the case of community catering (Table 1).

**Table 1.** Elements of Community Catering Socio-technical system.

| System Elements | Community Catering |
|---|---|
| Values | sustainability as value, regional products as value, meat quality, animal welfare, healthy nutrition |
| Knowledge | nutrition education, knowledge about healthy eating, recommendations for healthy nutrition (e.g., from the German Nutrition Society), environmental education |
| Behavior | menu choice, eating habits, trends towards health and sustainability, individualisation of nutrition |
| Products | complete menus, take-away offers, salad bar |
| Production | production processes in the canteen kitchen and in the upstream life cycle stages of the food (e.g., production of partially processed ingredients, such as vegetable broth, agricultural production) |
| Markets | catering in the education and training sector, catering in social institutions and clinics, catering in companies |
| Infrastructure | equipment of (large) kitchens, logistics of food distribution |
| Social structures | residence-workplace distances, family and work (time) models |
| Policy | hygiene regulation, expansion of day-care places and all-day schools, procurement guidelines |

## 3. Materials and Methods

In the specific case, an analysis of the status quo of the socio-technical system (Figure 3) of community catering in Germany was carried out as the basis for the application of the TRAFO 3.0 governance model. This analysis was based on data collected through multiple methods including a literature and qualitative desk research, as well as semi-structured interviews conducted with actors from the different areas of community catering (care, business, institutional and education) aiming to gather information on perceptions, experiences and strategies as well as limitations regarding meat reduction. The detailed semi-structured interviews were conducted during the second trimester of 2016 with a total of 8 respondents. Additionally, a workshop with the participation of 25 actors representing kitchen managerial staff and decision makers in community catering from all over Germany was held at the end of June 2016. This event was organized with the purpose of gathering further input and triangulating preliminary results derived from data collected through the other two methods.

For the analysis of this empirical data, the TRAFO 3.0 model was used to derive concrete governance approaches resulting from the different elements of the socio-technical system which were defined for this case study (Table 1). Building on these elements, the analysis was oriented to derive implications for the promotion of sustainability transformations in food systems with the aim of reducing the use of meat and other food products of animal origin in community catering. This means that the approaches described in Jacob et al. [24] for a transformative environmental policy were considered and examined together with classical policy instruments in order to show the possibilities for promoting and shaping the desired transformation.

## 4. Results of the Elements of Systemic Analysis of the Community Catering Socio-Technical System

This paper reports on empirical findings about innovations from actors of the community catering market. These findings resulted from applying the described governance model (Figure 3) to identify best-practices regarding reduction in meat consumption and derive implications for transformative environmental policy. The results contribute to define policy approaches for governing sustainability transitions in established food system as selected transformation domain.

### 4.1. Values

There is currently no common model for sustainable out-of-home catering or sustainable community catering that offers orientation and legitimacy for stakeholder action. The central stakeholder in community catering is the German Hotel and Restaurant Association (DEHOGA), in which the companies of the system and community catering are also represented. Except for the goal of reducing food waste, DEHOGA—according to individual companies—does not yet pursue a sustainability-related goal or mission statement. In particular, the reduction of the use of food of animal origin is not addressed by the association. However, it can be stated that best practice stakeholders from the various fields of community gastronomy, scientific stakeholders and various municipal stakeholders have identical ideas regarding the transformation goals of sustainable community catering [26,27]. The ecological, health and economic significance of a reduced use of animal products is undisputed among these actors.

Some currently observed trends and value developments in society are also currently having a beneficial effect on a transformation in the desired direction. Based on a literature review and analysis, Göbel et al. identified amongst others the megatrend "health" [28]. The megatrend "health" leads to an increasing health awareness permeating all areas of life and consumption behavior, including and above all in the field of nutrition. It is now undisputed that a high consumption of meat products goes hand in hand with a high health risk. In addition, "*health as one of the most important values of the 21st century will continue to gain importance*" [29]. A further trend that can be observed and that is conducive to transformation is the megatrend "sustainability" [28]. In the area of nutrition, it is mostly reflected in an increasing awareness of precarious forms of animal husbandry and the demand for more animal welfare as well as an increasing awareness of the ecological and social effects of global value chains in the food sector and resulting in increased demand for regional products.

### 4.2. Knowledge

The importance of reducing average meat consumption for health has long been reflected in the recommendations of the German Society for Nutrition and is also communicated by other stakeholders in the health sector, such as health insurance funds, consumer centers or various institutions in the field of nutrition counselling and nutrition education. The environmental relief effect of a dietary style that is characterized by a high proportion of plant products is also undisputed and has, for example, found its way into the 2016 climate protection report for the Federal Ministry of Food and Agriculture (BMEL) [30]. A number of best-practice stakeholders in the community catering sector as well as some cities and municipalities have already taken up the implementation of these recommendations before the climate protection report was published. This is because it has been recognized that a reduction in the quantities of meat in community catering can also be used to compensate for additional costs arising from the switch or partial switch to organic food [26]. Moreover, the aspect that inhibits transformation is that the preparation of low meat dishes is generally not sufficiently addressed in the training of specialists in the catering sector. All the interviewed kitchen managerial staff persons and decision makers in community catering underscored the importance of this statement. This represents a mature existing knowledge gap in the out-of-home catering industry.

### 4.3. Behavior

Especially among young adults and women, as well as in social groups with a high level of education, a change in dietary behavior towards a vegetarian or vegan diet can be observed [31]. This dietary behavior change is especially relevant for the Studierendenwerke (student unions), which normally coordinate university cafeterias and canteens in Germany. The interviewed central purchasing manager of one of the German Studierendenwerke explained that his institution as well as other Studierendenwerke have therefore been adapting their product range accordingly for several years now. According to a study conducted in 2016 by the Association for Consumer Research (GFK—Gesellschaft für Konsumforschung), however, the so-called flexitarians are also on the advance: "*Meanwhile the proportion of those who do not completely do without meat, but deliberately want to reduce meat consumption, is 37 percent*" [32].

How the individual's choice of food can be influenced in community catering has now been relatively well investigated. It has been shown that improved recipes, bonus systems and emotionally conveyed information (e.g., storytelling) are favored as intervention methods by consumers as well as by community catering stakeholders [33].

### 4.4. Products and Production

A central challenge that must be considered during the transformation at product level is that in aiming to reduce meat quantities, the design of menus must be fundamentally changed. This statement has been made by all interviewed kitchen staff and decision makers. One central result of the interviews was that vegetarian and low-meat dishes must taste good and be accepted as wholesome meals. The interview partners explained that this central fact has several implications and effects, as described in the following.

Attempts to halve the size of the Schnitzel and to double the amount of potato salad, or to replace the meat with soy products, for example, are generally not well received by customers and visitors. In technical terminology, it is said that the classic three-component structure of menus —meat, satiating side dish and vegetables/salad—must be abolished. The dishes are designed in such a way (e.g., meat as topping or three small sausages instead of one large one) that the guest does not even notice that he is eating less meat. However, kitchen managers and the executive staff often face challenges regarding how to proceed with menu planning within the framework of such a transformation, so that the customer is satisfied. This is due to the fact that the preparation of tasty vegetarian and vegan dishes, or dishes with a low proportion of ingredients of animal origin, are not commonly included in the training programs for professions in the catering sector. This is an extremely important transformation inhibiting factor both from the production perspective as well as from the knowledge as already mentioned. Many companies that set out to implement these measures therefore organize their own "further training events" for their kitchen teams. Guests can best be won over by taste and creativity for vegan, vegetarian or low meat dishes. Vegan or vegetarian action days are usually perceived as paternalism and are counterproductive with regard to the desired transformation.

### 4.5. Markets and Societal Structures

As a market for food, the importance of community catering has steadily increased in recent years due to social changes in work and family organization (including a steady increase in commuters and travel distances between home and work, an increase in full-day care for children and young people, an increase in the number of double earners households, an increase in the number of single households). Like the entire market for out-of-home catering, this sub-sector is in a growth phase [28]. In particular, community catering in day-care centers and schools is showing strong growth. According to the GFK [34], only "*just over one in two six- to nine-year-olds still have a midday meal at home on weekdays; for children between the ages of three and five it is even less. Compared to 2005, this is a decrease of 33 and*

*41 percent respectively*". Nevertheless, there is great price pressure in the industry and a very high proportion of small and micro enterprises [28].

### 4.6. Policy

Agricultural policy in recent years has basically had an inhibiting effect on transformation with regard to the transformation goal. The state framework conditions are designed in such a way that livestock farming throughout Germany is geared to resource efficiency. The Scientific Advisory Board on Agricultural Policy at the Federal Ministry of Food and Agriculture (BMEL) [35] has developed comprehensive recommendations for a strategy regarding a change in livestock husbandry, including concrete measures that also include changes in consumer habits. However, the implementation of the proposed measures requires a policy strategy coordinated across all ministries. For the time being, no efforts in this direction are discernible from the Federal Government.

In this regard, the National Program for Sustainable Consumption does not mention "reducing meat consumption" or "reducing the use of animal products" as concrete goals. Meanwhile the Federal Ministry for the Environment, Nature Conservation and Nuclear Safety has indicated that no meat or fish products should be offered at events organized by the Ministry. Nevertheless, so far, the reduction of the use of animal products in community catering has not been addressed through model invitations to tenders or procurement recommendations at federal level. Corresponding recommendations for the formulation of a requirement within the scope of the tender specifications for the award of concession contracts on the operation of canteens or for the award of catering services can be found in the practical guide "Mehr Bio für Kommunen" ("More organic products for municipalities") [26], which was prepared on behalf of the German Bio-Cities-Network.

At EU level, the recommendations for sustainable public procurement of food and catering services were being revised at the time of the study. The current recommendation also contains a point for a minimum requirement aimed at reducing the proportion of meat in the supply of community caterers and increasing the proportion of ingredients of plant origin [36]. Individual federal states and municipalities, such as the Saarland or Berlin require compliance with the corresponding recommendations of the German Nutrition Society as a mandatory basis for day-care and school catering in public tenders.

According to the results of Kimmons et al., such recommendations or guidelines as described above have a great potential to improve the health and sustainability of a specific food system [37].

## 5. Discussion

The results are discussed against the background of the concept of transformative environmental policy [24] described in Section 2.2. In the following we summarize those approaches which, on the basis of the initial analysis, have been identified as the relevant courses of action to promote transformation pathways to sustainable community catering.

### 5.1. Exploiting Societal Trends

The first initiatives towards a transformation towards sustainable community catering, including a reduction in the proportion of food of animal origin, have been launched in the last ten years. Interviewed best practice stakeholders explained that they reduce the use of meat in community catering, increase the attractiveness of vegetarian and vegan menus and use high-quality meat, often from regional production. They thus pick up on the prevailing social trends (health, animal welfare, regionality) and have been successful in this respect.

It is extremely important that the trends of animal welfare and regionality are adequately taken up by politicians. For example, consumers—with the exception of products from controlled organic animal husbandry—have not yet been able to clearly identify animal products that have been bred in farms that meet a higher animal welfare standard in the market. Similarly, the establishment and

expansion of regional value chains has so far been in the hands of individual stakeholders and is not supported on a higher level.

In addition, appropriate decisions or declarations of intent in the direction of reducing animal products as well as the attached measures, should also be accompanied by social marketing campaigns using storytelling approaches. Prominent authentic role models, for example from sport, who have changed their diet also with regard to their performance, should be won over for these marketing campaigns.

This aspect is supported by recent research results from the field of consumption behavior suggesting that social norms interventions are not universally applicable to promote behavior change [38]. In this sense, descriptive norm messages can produce boomerang effects, which is why the use of storytelling for a desired transformation should focus on simple messages in the context of current social trends.

*5.2. Networking Important Stakeholders (Including Change Agents)*

The knowledge about a sustainable design of out-of-home catering is already relatively broad. Particularly in the field of public procurement, valuable practical knowledge has been gathered over the past ten years for implementation. This means that in principle it is known how to deal with the challenges and obstacles that the reduction of the use of food of animal origin can pose, and despite this, it is possible to be successful on the market. However, there is still a lack of implementation in many areas. Here the networking of relevant stakeholders via appropriate regional and local platforms and/or food councils must be supported. In addition, networking with stakeholders from best-practice examples should be promoted. Suggestions and advice for implementation assistance that could be gained from supraregional success stories can accelerate the transformation. What is important here is the practical transfer of knowledge with regard to the experience gained in the preparation of attractive, low meat dishes that convince the guest with their taste.

All of this is aligned with the recognized relevance of social networks in the field of sustainability transition in facilitating information flows, spreading actors' visions as well as determining opportunities for action and roles taken by different stakeholders [39]. In this sense, the network structures in which stakeholders organize themselves are key in promoting opportunities to become change agents by taking the role of implementers and setting example practices which derive in power to the influence outcomes in a transition process.

In addition, new alliances should be forged and new stakeholders integrated, or actors with the same interests should be better networked. The topic could, for example, also be brought to a wide variety of companies via various business associations or training courses for work councils under the heading of "company health management". If necessary, cooperation could also be established with health insurance funds.

In the area of day-care and school catering, for example, the committee of the Federal States' Parents' Advisory Council and the respective state ministries responsible for education, as well as other regional educational actors, should be mentioned.

It is urgently necessary that the preparation of tasty vegetarian and vegan dishes, or the preparation of dishes with a low proportion of ingredients of animal origin, becomes an integral part of the various gastronomic training courses both in theory and practice. The lack of knowledge about the preparation of target group-oriented tasty alternatives represents a major obstacle to a transformation towards increasing the proportion of plant-based foods in out-of-home catering. Here the Federal Institute for Vocational Education and Training (BIBB), as the competent authority for the nationwide training regulations and the nationwide training framework plans, and the German Hotel and Restaurant Association (DEHOGA) play a central role.

### 5.3. Promoting (Also Non-Technical) Innovations and Design Exnovation and System Interfaces

Taking into consideration the existence of established best-practice examples, it is important within the framework of the desired transformation that the supply of dishes containing only a low proportion of animal products and the consumption of these dishes in the context of community catering are increased.

As already mentioned above, the fact that the necessary practical knowledge for the preparation of this type of dishes is not widespread in the catering sector, is an obstacle in following a transformation pathway. The transformation can only succeed if employees in the out-of-home catering sector are able to prepare tasty, creative and nutritionally complete low-meat or plant-based meals. Good staff training is a key factor for the desired transformation [40]. According to Tsui and Morillo, cooks are playing a central role in translating nutritional guidelines like recommendations for lower meat consumption in tasty and well-accepted recipes [41]. It should also be noted that different menu lines have to be developed for different target groups (seniors, toddlers, young people), which consider the taste preferences of the respective target group.

In this context, introducing changes in the product range available in the community catering providers could be seen as a nudging strategy for promoting healthier and sustainable nutrition behavior. Such nudging strategies have already been proved positive results in the topic of meat reduction [42] and can thus be considered effective alternatives to the use of descriptive social norms, such as the attempt for "veggie Day" in Germany which have failed in addressing this social challenge [12].

Until the theoretical and practical content of the training programs for catering professions has been revised and the next generation has been trained, it will be necessary for the kitchen teams of community catering establishments to receive additional training in order to gain these skills and knowledge. Practice stakeholders interviewed as part of the TRAFO 3.0 research project have also pointed out the great importance of so-called "caretakers" for a successful transformation. In general, it is extremely important that an appropriate person takes charge of the implementation in the individual case. This person must also bear in mind that the entire team must be involved in this process and that its performance has to be rewarded—at least verbally.

In accordance, suitable initiatives, measures and approaches should be developed and launched so that a corresponding value shift in society in the field of "nutrition" (e.g., through appropriate information and education campaigns or by including the topic in the educational mandate of day-care centers and schools) can be widely initiated and supported. This can also be promoted through the use of interfaces to other systems such as health and work council activities in the business and institutional contexts. In order to reach target groups that have not yet been reached, new approaches should be tested, such as the use of social networks or suitable gamification approaches. In parallel, suitable intervention measures can be used directly in the specific catering situation to support and accelerate the desired transformation [33].

Another aspect to consider is that the services from the community catering sector are of relatively little value to the majority of customers. According to the interviewed partners, the willingness to pay for food offered by community catering actors is relatively low (mainly in day-care and school catering sectors). This aspect is particularly relevant in view of the fact that the high price pressure that exists in the out-of-home catering markets, especially affecting the community catering sector, is an important factor that inhibits transformation. A supportive effect could be achieved here by sets of measures involving an increase in the price of food products from animal origin. In the context of the implementation of these measures, however, accompanying strategies should also be adopted to mitigate possible social impacts. Such mitigation strategies for example, could involve municipal subsidies for day-care and school meals for children from low-income backgrounds.

Last but not least, a clear joint commitment to a healthy, low meat diet by the relevant federal ministries could support the transformation and also drive it forward through the market and

purchasing power of the public sector. Ideally, this could be regulated in a binding "joint decree" comparable to the one existing for the procurement of wood products.

In this regard, it is relevant to consider that for the field of for food democracy, state actors have identified to have key roles in related participation processes as potential initiators, shapers and implementers [43]. The level of engagement of the governmental sector depends on how they interact with actors of local food markets will determine governance for food system transformation processes.

### 5.4. Thinking Along with Production Requirements

The sustained high production of animal products in Germany, which is oriented towards further increases in efficiency and the expansion of export activities, is currently leading to growing environmental pollution, deficits in animal welfare, low producer prices and the destruction of farming structures associated with the latter (number of farms being decimated). The reductions in the consumption of animal products through innovations introduced by community catering stakeholder as presented here have a clear potential for the health benefits of the population. However, despite the fact that these innovations represent exemplary practices in global terms, they do not yet lead to alleviation for the environment and animals.

A major support for a transformation of our food system, including the reduction of animal products in out-of-home catering, could therefore be the development of an overarching political strategy (including an action program) towards a "sustainable agricultural production and nutrition systems" along the lines of the French model. This strategy should have a broad foundational structure and therefore be developed with wide participation not only at ministerial level, but also with the involvement of relevant social stakeholders. At this point, it has to be pointed out that this strategy should also include elements for a sustainability transformation of the meat sector. Meat production is not unsustainable per se. Sustainable animal husbandry is an element of closed agricultural nutrient cycles. Furthermore, biodiversity benefits from extensive sustainable livestock systems, and last but not least, from a global perspective, sustainable livestock systems are necessary for global food security. The consumption of meat and other animal food products are not unsustainable per se. Sustainable produced animal food products can be part of a sustainable diet.

### 5.5. Policy Implications for Transformations in Food Systems

According to the applied governance model for examining changes in socio-technical systems, transformations represent not only changes to individual elements, but also to the entire system. Considering the results and implications discussed above, it is understood that innovations in the community catering sector, as part of the out-of-home catering market is serving as transformation realm for socio-ecological change [14].

The diversity of empirical findings resulting from this research project, offers additional evidence in support of the understanding of sustainability transitions as multi-dimensional processes. Furthermore, the variety of novel stakeholder approaches to the social challenge of meat reduction in community catering sector, reveals that regime actors are indeed actively involved in addressing the need for more sustainable food systems, and contributing to the emergence of new trajectories within several of the examined elements.

As regime transformations are characterized, according to Ghosh and Schot [21], by changes in three dimensions (out of five) as a threshold for transformation, these identifies trajectories should be carefully monitored and promoted in order to enable the successful development of this process. From the perspective of transformative environmental policy [24], the measures discussed above should accompany this process so that a change in regulative and normative rules is simultaneously promoted thereby, having a deeper impact on the current regime.

**Author Contributions:** Conceptualization, J.T. and V.L.; methodology, C.-O.G.; investigation, J.T.; writing—original draft preparation, J.T. and V.L.; writing—review and editing, J.T. and V.L.; project administration, C.-O.G.; funding acquisition, C.-O.G. All authors have read and agreed to the published version of the manuscript.

**Funding:** This research was funded by the German Federal Ministry of Education and Research (BMBF, funding number: 01UT1426) and by the German Federal Environmental Agency (UBA, funding number: 3717 11 101 1).

**Conflicts of Interest:** The authors declare no conflict of interest.

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
