# Peer review of "How a Transformation towards Sustainable Community Catering Can Succeed"

_sustainability, doi:10.3390/su12010101_

Round 1
Reviewer 1 Report
Thank you for the opportunity to review this paper. Overall I find it a clean study that takes a creative and theoretically-informed approach to how meat consumption can be reduced in community foodservice settings.
Overall I recommend publication with minor revisions but have some suggestions:
A wider array of English-language literature will be made apparent by switching keywords from "community catering" to "institutional foodservice." See attached article, for example. While reference is made to qualitative interviews, no quotes are used to inform the analysis. Please provide quotations and data presentation from interviews as I suspect that will make the article more interesting and give credence to the secondary analysis. Some explanation of TRAFO 3.0 is needed for the uninitiated. It is presented as if the reader will be familiar with the project. Line 263: is health really a megatrend? Or a key facet of life. Some nuance/citation is needed. In discussion: obviously the meat industry--farmers, workers, investors, etc. have much to lose here. More of that should be discussed as well as sustainable meat programs--it is not always so black and white as meat bad, veggie good as the recent flourishing of unhealthful meat alternatives has shown.
Author Response
Response to Reviewer 1 Comments, 2019-12-11
Point 1: A wider array of English-language literature will be made apparent by switching keywords from "community catering" to "institutional foodservice." See attached article, for example.
Response 1: We added the keyword “institutional foodservice” and introduced this keyword into the paper. We didn´t exchange the terms, since some literature will be found if you use the keyword “community catering”.
We also did an additional literature research with the keyword “institutional foodservice” and incorporated the following relevant additional references into the paper:
Kimmons, J.; Jones, S.; McPeak, H.H.; Bowden, B. Developing and implementing health and sustainability guidelines for institutional food service. Adv. Nutr. 2012, 3, 337–342, doi:10.3945/an.111.001354. (page 10) Hamerschlag, K.; Kraus-Polk, J. Shrinking the Carbon and Water Footprint of School Food: A recipe for combating climate change. A pilot analysis of Oakland Unified School District’s Food Programs, 2017. Available online: https://www.heyheyrenee.com/wp-content/uploads/2017/02/FOE_FoodPrintReport_7F.pdf. (page 11) Tsui, E.K.; Morillo, A. How cooks navigate nutrition, hunger and care in public-sector foodservice settings. Public Health Nutr. 2016, 19, 946–954, doi:10.1017/S1368980015002086. (page 11)
Point 2: While reference is made to qualitative interviews, no quotes are used to inform the analysis. Please provide quotations and data presentation from interviews as I suspect that will make the article more interesting and give credence to the secondary analysis
Response 2:
We added several quotations from the interviews. Please note:
Chapter 4.2, lines 304, 305: This statement was emphasized by all the interviewed kitchen managerial staff persons and decision makers in community catering. Chapter 4.3, line 310-314: Especially among young adults and women, as well as in social groups with a high level of education, a change in dietary behaviour towards a vegetarian or vegan diet can be observed [31]. This dietary behaviour change is especially relevant for the Studierendenwerke (student unions), which normally coordinate university cafeterias and canteens in Germany. The interviewed central purchasing manager of one of the German Studierendenwerke explained that his institution as well as other Studierendenwerke have therefore been adapting their product range accordingly for several years now. Chapter 4.4, line 324-329: A central challenge that must be taken into account during the transformation at product level is that in aiming to reduce meat quantities, the design of menus must be fundamentally changed. This statement has been made by all interviewed kitchen staff and decision makers. One central result of the interviews was that vegetarian and low-meat dishes must taste good and be accepted as wholesome meals. The interview partners explained that this central fact has several implications and effects, as described in the following. Attempts to halve the size of the Schnitzel and to double the amount of potato salad, or….. Chapter 5.1, line 391-396: The first initiatives towards a transformation towards sustainable community catering, including a reduction in the proportion of food of animal origin, have been launched in the last ten years. Interviewed best practice stakeholders explained that they, for example, reduce the use of meat in community catering, increase the attractiveness of vegetarian and vegan menus and use high-quality meat, often from regional production. They thus pick up on the prevailing social trends (health, animal welfare, regionality) and have been successful in this respect. Chapter 5.3, line 468-472: Practice stakeholders interviewed as part of the TRAFO 3.0 research project have also pointed out the great importance of so-called “caretakers” for a successful transformation. In general, it is extremely important that an appropriate person takes charge of the implementation in the individual case. This person must also bear in mind that the entire team must be involved in this process and that its performance has to be rewarded – at least verbally. Chapter 5.3, line 482-292: According to the interviewed partners, the willingness to pay for food offered by community catering actors is relatively low (mainly in day-care and school catering sectors). This aspect is particularly relevant in view of the fact that the high price pressure that exists in the out-of-home catering markets, especially affecting the community catering sector, is an important factor that inhibits transformation.
Point 3: Some explanation of TRAFO 3.0 is needed for the uninitiated. It is presented as if the reader will be familiar with the project.
Response 3: We added to the abstract following passage (see lines 77, 78): “…The challenges and obstacles associated with the transformation and the governance approaches they convey are being investigated within the framework of the BMBF-funded research project “Governance model for socio-ecological transformation processes in practice: development and testing in three areas of application” carrying the acronym TRAFO 3.0 (http://trafo-3-0.de).”.
Furthermore, we described the background, the overall approach and general outcomes of the project in chapter 2.2 (see line 144 to 160: “2.2. A governance model for socio-ecological transformation).
In order to shift the current energy and resource-intensive lifestyles of our society towards sustainable production and consumption patterns, a sustainability transformation is necessary. This involves a multi-faceted change in societal routines and structures, including the integration of social and technological aspects. Although several theories already exist on the emergence and process of transformations, it still remains unclear how exactly transformations can be strategically promoted, potentially initiated, and shaped. This was the point of departure of the research project TRAFO 3.0 founded by the Federal Ministry of Education and Research in Germany from April 2015 to March 2018.
Building on theoretical elements of transitions research and evolutionary economics, a draft governance model for examining socio-ecological transformation processes in practice (Figure 3) was developed in the context of the project TRAFO 3.0. This draft model was tested in three fields of application (paperless publishing and reading, wide use of electric bikes, sustainable production and consumption of meat) and the findings from the analysis of these three fields of application were incorporated in an iterative process into the final governance model. Furthermore, based on the results of the three fields of application, a set of approaches for the promotion and governance of small- and medium-range transformations was provided. …”
Point 4: Line 263: is health really a megatrend? Or a key facet of life. Some nuance/citation is needed.
Response 4:
In line 279-280 we added this citation: Some currently observed trends and value developments in society are also currently having a beneficial effect on a transformation in the desired direction. Based on a literature review and analysis, Göbel et al. identified amongst others the megatrend “health” [2]. Göbel, C.; Scheiper, M.-L.; Teitscheid, P.; Müller, V.; Friedrich, S.; Engelmann, T.; Neundorf, D.; Speck, M.; Rohn, H.; Langen, N. Nachhaltig Wirtschaften in der Außer-Haus-Gastronomie. Status-quo-Analyse - Struktur und wirtschaftliche Bedeutung, Nachhaltigkeitskommunikation, Trends. NAHGAST Arbeitspapier 1., 2017.
Point 5: In discussion: obviously the meat industry--farmers, workers, investors, etc. have much to lose here. More of that should be discussed as well as sustainable meat programs--it is not always so black and white as meat bad, veggie good as the recent flourishing of unhealthful meat alternatives has shown.
Response 5:
This statement above is correct, but it was not in the focus of our paper. We tried to address this point in chapter 5.4 by supplementing this passage: “A major support for a transformation of our food system, including the reduction of animal products in out-of-home catering, could therefore be the development of an overarching political strategy (including an action programme) towards a “sustainable agricultural production and nutrition systems” along the lines of the French model. This strategy should have a broad foundational structure and therefore be developed with wide participation not only at ministerial level, but also with the involvement of relevant social stakeholders. At this point it has to be pointed out that this strategy should also include elements for a sustainability transformation of the meat sector. Meat production is not unsustainable per se. Sustainable animal husbandry is an element of closed agricultural nutrient cycles. Furthermore, biodiversity benefits from extensive sustainable livestock systems, and last but not least, from a global perspective, sustainable livestock systems are necessary for global food security. The consumption of meat and other animal food products are not unsustainable per se. Sustainable produced animal food products can be part of a sustainable diet.”
Reviewer 2 Report
Minor suggestions:
Keywords: they are not keywords, they are sentences, suggestion to be more concise.
References:10 references in German, difficult to the majority of readers to undesrtand.
Author Response
Response to Reviewer 2 Comments, 2019-12-11
Point 1: Keywords: they are not keywords, they are sentences, suggestion to be more concise.
Response 1:
We changed some of the keywords: community catering; institutional foodservice; sustainability transformation; transformation governance; sustainable food systems; meat reduction; menu planning; social innovations; environmental policy
Point 2: References: 10 references in German, difficult to the majority of readers to undestand
Response 2:
I understand this objection. We supplemented few relevant references in English (see below). The problem is that we analysed the German Community Catering Socio-technical system. Therefore, some relevant literature was only available in German. We couldn´t replace it by an English paper/document. See for example: Weingarten, P.; Bauhus, J.; Arens-Azevedo, U.; Balmann, A.; Biesalski, H.K.; Birner, R.; Bitter, A.; Bokelmann, W.; Bolte, A.; Bösch, M.; et al. Klimaschutz in der Land- und Forstwirtschaft sowie den nachgelagerten Bereichen Ernährung und Holzverwendung: Gutachten des Wissenschaftlichen Beirats für Agrarpolitik, Ernährung und gesundheitlichen Verbraucherschutz und des Wissenschaftlichen Beirats für Waldpolitik beim Bundesministerium für Ernährung und Landwirtschaft. Berichte über Landwirtschaft – Zeitschrift für Agrarpolitik und Landwirtschaft 2016, 1–479. Or the current results of the research project “NAHGAST” (https://www.nahgast.de/) founded by the Federal Ministry of Education and Research in Germany from April 2015 to March 2018, that were quite relevant for the paper.
Supplemented references:
Kimmons, J.; Jones, S.; McPeak, H.H.; Bowden, B. Developing and implementing health and sustainability guidelines for institutional food service. Adv. Nutr. 2012, 3, 337–342, doi:10.3945/an.111.001354. Hamerschlag, K.; Kraus-Polk, J. Shrinking the Carbon and Water Footprint of School Food: A recipe for combating climate change. A pilot analysis of Oakland Unified School District’s Food Programs, 2017. Available online: https://www.heyheyrenee.com/wp-content/uploads/2017/02/FOE_FoodPrintReport_7F.pdf. Tsui, E.K.; Morillo, A. How cooks navigate nutrition, hunger and care in public-sector foodservice settings. Public Health Nutr. 2016, 19, 946–954, doi:10.1017/S1368980015002086.